# Structural heterogeneity of cellular K5/K14 filaments as revealed by cryo-electron microscopy

**Miriam S Weber[1], Matthias Eibauer[1], Suganya Sivagurunathan[2], Thomas M Magin[3], Robert D Goldman[2], Ohad Medalia[1]\***

[1]Department of Biochemistry, University of Zurich, Zurich, Switzerland; [2]Department of Cell and Developmental Biology, Northwestern University Feinberg School of Medicine, Chicago, United States; [3]Institute of Biology, University of Leipzig, Leipzig, Germany

**Abstract** Keratin intermediate filaments are an essential and major component of the cytoskeleton in epithelial cells. They form a stable yet dynamic filamentous network extending from the nucleus to the cell periphery, which provides resistance to mechanical stresses. Mutations in keratin genes are related to a variety of epithelial tissue diseases. Despite their importance, the molecular structure of keratin filaments remains largely unknown. In this study, we analyzed the structure of keratin 5/keratin 14 filaments within ghost mouse keratinocytes by cryo-electron microscopy and cryo-electron tomography. By averaging a large number of keratin segments, we have gained insights into the helical architecture of the filaments. Two-dimensional classification revealed profound variations in the diameter of keratin filaments and their subunit organization. Computational reconstitution of filaments of substantial length uncovered a high degree of internal heterogeneity along single filaments, which can contain regions of helical symmetry, regions with less symmetry and regions with significant diameter fluctuations. Cross-section views of filaments revealed that keratins form hollow cylinders consisting of multiple protofilaments, with an electron dense core located in the center of the filament. These findings shed light on the complex and remarkable heterogenic architecture of keratin filaments, suggesting that they are highly flexible, dynamic cytoskeletal structures.

**\*For correspondence:** omedalia@bioc.uzh.ch

**Competing interests:** The authors declare that no competing interests exist.

## Introduction

Keratin Intermediate Filaments (KIFs) are an essential component of the cytoskeleton of epithelial cells. KIFs are classified as type I and type II Intermediate Filament (IF) proteins, according to their sequence (*Szeverenyi et al., 2008*; *Bragulla and Homberger, 2009*; *Toivola et al., 2015*). Keratins form a highly flexible and dynamic filamentous network in the cytoplasm (*Etienne-Manneville, 2018*; *Pora et al., 2020*; *Kölsch et al., 2010*; *Robert et al., 2016*; *Windoffer et al., 2004*; *Yoon et al., 2001*). Their main known function is to protect the cell from external stresses by providing mechanical stability and ensuring the integrity of tissues through cell-cell and cell-matrix contacts. Point mutations in keratin genes are associated with cell and tissue instabilities and severe diseases, termed keratinopathies (*Toivola et al., 2015*; *Haines and Lane, 2012*). For example, the skin blistering disease Epidermolysis Bullosa Simplex (EBS) is caused by point mutations in the Keratin 5 (K5) and 14 (K14) genes (*Jacob et al., 2018*; *Coulombe et al., 1991*; *Coulombe and Lee, 2012*). A key to understanding the function of KIFs in both normal and diseased cells is to unveil the structural organization of the filaments.

Keratin proteins are composed of three domains: a highly conserved α-helical central rod domain, known to facilitate filament assembly, and intrinsically disordered head and tail domains. The latter

are highly post-translationally modified and are regarded as essential for regulatory functions and filament stability (*Wilson et al., 1992*; *Parry and Steinert, 1999*; *Chernyatina et al., 2015*). Keratins are obligatory heterodimers, formed by the parallel and in-register assembly of an acidic type I and a basic type II keratin protein. Two dimers assemble into antiparallel tetramers, which can further align laterally and longitudinally to build mature keratin filaments (*Steinert et al., 1993*; *Lee et al., 2020*; *Herrmann and Aebi, 2016*). Longitudinally elongated tandem arrays of tetrameric subunits are termed protofilaments.

Although KIFs have been studied intensely for many years, details of their molecular architecture remain largely unknown (*Eldirany et al., 2021*). On the level of keratin monomers and dimers, crystallographic studies have provided high-resolution insights into the organization of small regions of the central rod domain (*Lee et al., 2020*; *Lee et al., 2012*; *Bunick and Milstone, 2017*; *Eldirany et al., 2019*) and molecular dynamics simulations have presented a 3D model of a complete K1/K10 dimer (*Bray et al., 2015*). Four different modes of tetramer assembly have been identified by cross-linking studies, describing how higher order oligomers form during filament assembly (*Steinert et al., 1993*). It is expected that filament assembly starts by the formation of $A_{11}$ tetramers, where the 1B domains of the rod of two adjacent dimers interact in a half-staggered, antiparallel arrangement (*Lee et al., 2020*; *Herrmann and Aebi, 2016*). Tetramers can elongate longitudinally by formation of $A_{22}$ interactions, where the 2B domains of adjacent rods overlap (*Lee et al., 2020*; *Herrmann and Aebi, 2016*). Laterally, neighboring tetramers interact via $A_{12}$ bindings to form 10 nm wide filaments (*Lee et al., 2020*; *Herrmann and Aebi, 2016*). However, little is known about the 3D high-resolution structure of mature keratin filaments. It is generally accepted that keratin filaments are helical assemblies consisting of multiple protofilaments (*Astbury, 1939*; *Crick, 1952*; *Crick, 1953*), which form a cylindrical tube (*Parry and Steinert, 1999*). However, the exact number of protofilaments per filament and therefore the number of keratin monomers per cross section is still debated and may vary with respect to the specific type I/type II keratin pairs expressed. Mass-Per-unit-Length (MPL) measurements of recombinant in vitro assembled K8/K18 filaments have suggested that they are built from 16 to 25 monomeric protein chains in cross section, depending on the ionic strength of the buffer and the assembly time (*Herrmann et al., 1999*). Interestingly, MPL measurements of KIFs assembled in vitro from keratins extracted from human epidermis have revealed that the majority of them contain 13–16 polypeptides in cross section, with fewer numbers of KIF comprised of either 20–26 or 28–35 polypeptides (*Engel et al., 1985*). These variations have been attributed to structural polymorphism of KIFs and apparently occur by varying the number of protofilaments along and among keratin filaments (*Engel et al., 1985*). In addition, there is a lack of consensus as to whether KIFs are hollow or filled tubes and whether they contain an internal structure (*Parry and Steinert, 1999*; *Bruce Fraser et al., 2003*; *Parry, 1996*; *Fraser and Parry, 2017*; *Watts et al., 2002*; *Herrmann and Aebi, 2004*).

Here, we studied the structure of native cellular K5/K14 filaments by Cryo-Electron Microscopy (cryo-EM) and Cryo-Electron Tomography (cryo-ET) (*Weber et al., 2019*). Since the expression of keratin isoforms is variable and complex in cultured cells, we prepared a cell line expressing filaments composed of K5/K14 only and studied their architecture within cells that were grown and lysed on EM grids, that is ghost cells. This process avoids potential structural artifacts due to in vitro assembly of KIFs. Our cryo-EM analysis revealed the remarkably heterogenic nature of keratin filaments and uncovered changes in the diameter and the helical pattern propagating along the filament. Cryo-ET and analysis of filament cross-sections revealed that the K5/K14 filaments are composed of a hollow cylinder with an internal electron dense core. The wall of the cylinder is constructed of a ring of six protofilaments. Our results quantify the flexibility of keratin filaments and uncover the immense structural heterogeneity of individual K5/K14 filaments.

## Results

### Generation of mouse keratinocytes expressing only K5/K14 filaments

Heterogeneity is a challenging problem that hampers structural determination (*Scheres, 2016*). Thus, the occurrence of multiple keratin pairs in most epithelial cells hinders the structural analysis of keratin filaments in their native environment, due to intrinsic structural heterogeneity (*Moll et al., 2008*). In this study, we set to gain insights into the architecture of cellular K5/K14 filaments.

Therefore, we utilized the murine keratinocyte cell line KtyI KO K14, which expresses the K5, K6 and K14 proteins as their only keratin isoforms, and the only IF proteins (*Homberg et al., 2015*; *Kumar et al., 2015*; *Kumar et al., 2016*). To reduce the keratin expression to K5 and K14 only, CRISPR/Cas9 was used to knock out the *Krt6a* and the *Krt6b* gene, which share 92.6% sequence identity (Materials and methods section). Although a small amount of wild-type *Krt6b* DNA was retained (*Figure 1—figure supplement 1A,B*), immunostaining revealed that no filaments containing K6 assemble in the resulting K5/14_1 cell line, and therefore do not affect the structural analysis carried out in our study (*Figure 1—figure supplement 1C*). We therefore conclude that the keratin filaments in this cell line consist only of K5/K14 protein pairs. A careful analysis of the KIF network after the CRISPR/Cas9 knockout procedure by confocal fluorescence microscopy indicated no obvious impact on the K5/K14 filament network (*Figure 1A*, *Figure 1—figure supplement 1C*).

## Cellular keratin filaments revealed by cryo-electron microscopy

K5/14_1 cells were cultured on cryo-EM grids and subjected to cytoskeleton extraction buffer that permeabilizes the cells and removes soluble cytoplasmic components and nuclear structures, producing IF-enriched ghost cells (Material and Methods section) (*Hu et al., 2019*; *Turgay et al., 2017*; *Kronenberg-Tenga et al., 2021*; *Svitkina and Borisy, 1998*; *Svitkina et al., 1995*; *Sailer et al., 2010*). The ghost cells were instantly plunge frozen and imaged by cryo-EM and cryo-ET (*Figure 1B–D*). Keratin filaments could be easily identified in cryo-EM micrographs (*Figure 1C*, blue arrows, *Figure 1—figure supplement 1D*), while the 3D organization of the keratins within the ghost cells was revealed by cryo-ET (*Figure 1D* light blue, *Video 1*). Actin filaments were detected in the sample as well (*Figure 1C*, orange arrows and D, red, *Figure 1—figure supplement 1D*) and used as an internal quality control for structural preservation by the extraction protocol and for cryo-EM image quality. Under these conditions, the structure of cellular F-actin was resolved to 6.1 Å using single particle image processing (*Figure 1—figure supplement 2*). Moreover, images were acquired from variety of cellular positions, encompassing internal to peripheral regions to ensure that the keratin IF analysis is not biased by cellular location.

Keratins form a complex filamentous network, including thick bundles containing numerous filaments, meshworks, in which filaments are often crossing and interacting with each other, as well as long stretches of individual filaments (*Figure 1C,E*, *Figure 1—figure supplement 1D*). Keratin filaments exhibit a wide range of shapes suggesting a high degree of flexibility. While some filaments are straight over long distances (several hundreds of nm), others exhibit a wavy appearance (*Figure 1E*, arrow). Additionally, highly bent keratin filaments are frequently detected (*Figure 1F*, *Figure 1—figure supplement 1D*). We traced 65 of these highly bent filaments in cryo-EM micrographs and determined their minimal apparent persistence length (contour length upon 90° turn, *Figure 1H*). Keratin 5/14 filaments are able to undergo a 90° turn within 118.4 ± 39.2 nm (*Figure 1G*), similar to distances observed for nuclear lamins (*Turgay et al., 2017*; *Tenga and Medalia, 2020*). These measurements were conducted only on the sub-population of highly bent filaments and not on the full range of shapes detected (e.g. straight filaments), as their varying behavior prohibited us to describe all of them with a single measure (*Block et al., 2015*). Interestingly, some filaments undertake even 180° turns within a few hundred nanometers without breaking and change directions multiple times within the inspected field of view (*Figure 1F*). In agreement with previous in vitro assembled keratin analyses, the remarkable flexibility of keratin filaments supports their ability to maintain filament integrity even under extreme conditions (*Ma et al., 2001*; *Coulombe and Fuchs, 1990*; *Herrmann et al., 2002*; *Köster et al., 2015*).

## Heterogeneity in filament diameter and helical pattern

To obtain deeper insights into the architecture of keratin filaments, we extracted 55 nm long straight keratin segments picked along filaments in ~1700 cryo-EM micrographs and subjected them to structural averaging by single particle data processing (*Scheres, 2012*; *He and Scheres, 2017*). A helical pattern that spirals along the long filament axis can be detected in several class averaged structures (*Figure 2A*, *Figure 2—figure supplement 1A*). While several structural classes show a clear helical pattern, others reveal elongated, rather straight sub-structures without an apparent helical symmetry (*Figure 2A*, bottom, *Figure 2—figure supplement 1A*). Transition regions between the two distinguished patterns can also be detected (*Figure 2A*, arrows). The mean diameter of

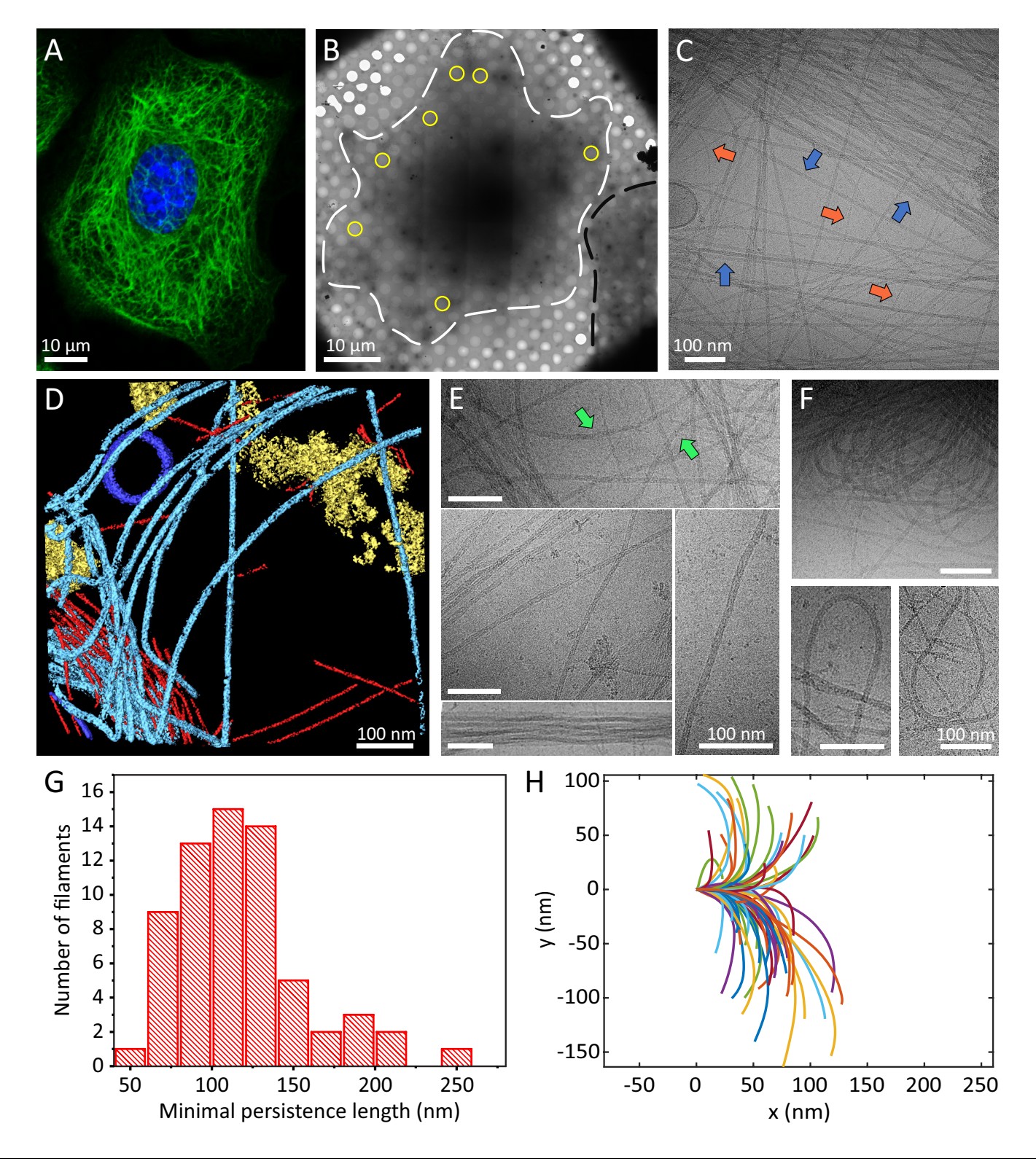

**Figure 1.** Cellular K5/K14 filaments as revealed by light and cryo-electron microscopy. (**A**) The murine keratinocyte cell line K5/K14_1 expressing only K5 and K14 filaments forms a complex KIF meshwork, as revealed by confocal immunofluorescence. Cells were stained for K14 (green) and chromatin (blue). (**B**) Ghost cells were analyzed by cryo-EM and cryo-ET. Low-magnification image of a cell grown on an EM-grid and treated with detergent prior to vitrification. Cell boundaries (dashed white line) are detected as well as a neighboring cell (dashed black line). Typical regions that were analyzed by

*Figure 1 continued on next page*

*Figure 1 continued*

cryo-EM are marked (yellow circles). (**C**) A typical cryo-EM micrograph of a ghost cell imaged at a higher magnification allows the detection of keratin filaments and other cytoskeletal elements (n=1860). Keratin filaments (blue arrows) and actin filaments (orange arrows) are distinguished by their characteristic diameter. A large keratin bundle is visible in the top right corner. (**D**) Surface rendering view of a cryo-tomogram of a ghost cell (n=44). Keratin filaments (light blue), actin filaments (red), vesicles (dark blue), and cellular debris (yellow) were manually segmented. (**E**) Different organizations of keratin filaments observed in the cryo-EM micrographs (n=1860), including straight filaments (middle), curved (top, green arrows) and bundled filaments (bottom left). Scale bars: 100 nm. (**F**) Highly bent keratin filaments are found within cryo-EM micrographs of ghost cells. Scale bars: 100 nm. (**G**) Quantification of the minimal apparent persistence length measurements performed on (n=65) highly bent keratin filaments extracted from cryo-EM micrographs. (**H**) A plot combining 65 contours of filaments that were used for the minimal apparent persistence length measurements in (**G**). Individual filaments, shown in different colors, are aligned at their origins for visualization purposes.

The online version of this article includes the following figure supplement(s) for figure 1:

**Figure supplement 1.** Knockout of K6 isoforms by CRISPR/Cas9.

**Figure supplement 2.** Validation that the sample preparation allows us to retrieve data at sub-nanometer resolution.

keratin filaments, as determined by direct measurement of intensity line-profiles through the class averages, is 10.1 ± 0.5 nm (*Figure 2B,C*), in agreement with previous observations (*Herrmann and Aebi, 2016*). A mean intensity line-profile through a lateral average of the most populated classes defined the edges of the filaments as well as a central density peak (*Figure 2C*). The outer boundaries of the filaments show the highest electron density values and therefore are their most pronounced structural features (i.e. the filament diameter), while a central density peak with slightly lower intensity is also apparent. This analysis further revealed a subset of structural classes with a much larger diameter than the majority of filaments (*Figure 2D*). Intensity line profiles of a thicker class (black asterisk) and a more frequently detected class (blue asterisk) indicate a 30% difference in filament diameter, 13.2 nm vs 10.1 nm, respectively (*Figure 2E*). Moreover, the internal structure of the thicker classes diverges from the classes shown in *Figure 2A*. Specifically, some classes reveal two distinct linear electron densities within the filament (*Figure 2D*, arrowheads), indicating a less dense packing of the individual protofilaments as compared to the compact classes. Others capture transitions between a thinner and a thicker region along an individual filament (*Figure 2D*, arrows). These findings indicate deviations in the organization of protofilaments, reflecting structural heterogeneity along individual filaments.

In order to determine the repeating unit, that is the pitch of the helical patterns observed in the class averages we calculated autocorrelation spectra for each helical class (*Figure 2F*, *Figure 2—figure supplement 1D*). Using this approach, the helical pitch can be determined, which is an intrinsic parameter of helical assemblies and reflects a 360° rotation of the helix (*Diaz et al., 2010*). We found that the pitch of the helical pattern varies dramatically among different classes, ranging from ~132 Å to ~163 Å (*Figure 2F*). Between these two extremes numerous distinct values for the pitch of the helical pattern can also be identified (*Figure 2—figure supplement 1D*). There are two possible scenarios that can explain a varying pitch distance: Either, a variable pitch is an intrinsic property of keratin filaments, which would add to the structural heterogeneity observed throughout this study, or the diverging pitch originates from filaments that lay tilted and not flat in the ice layer. Since keratin filaments are very flexible and form a 3D network in cells, we exploited the possibility that the different pitch lengths reflect filaments that are oriented out of plane. The projection of a tilted filament in our cryo-EM micrographs would therefore yield classes with an apparent shorter repeating pattern. In this case, the class with the longest

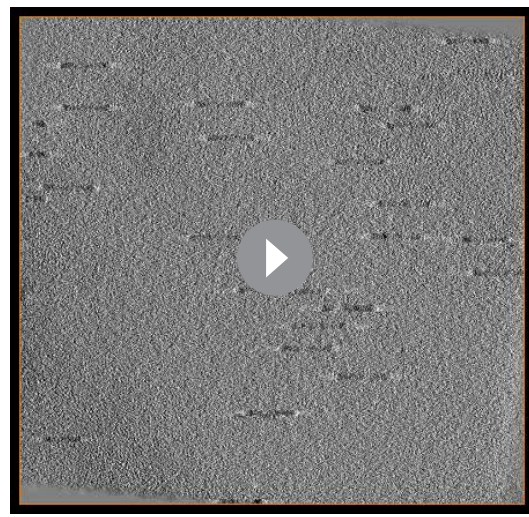

**Video 1.** Cryo-tomogram of a keratin network in a ghost cell.

https://elifesciences.org/articles/70307#video1

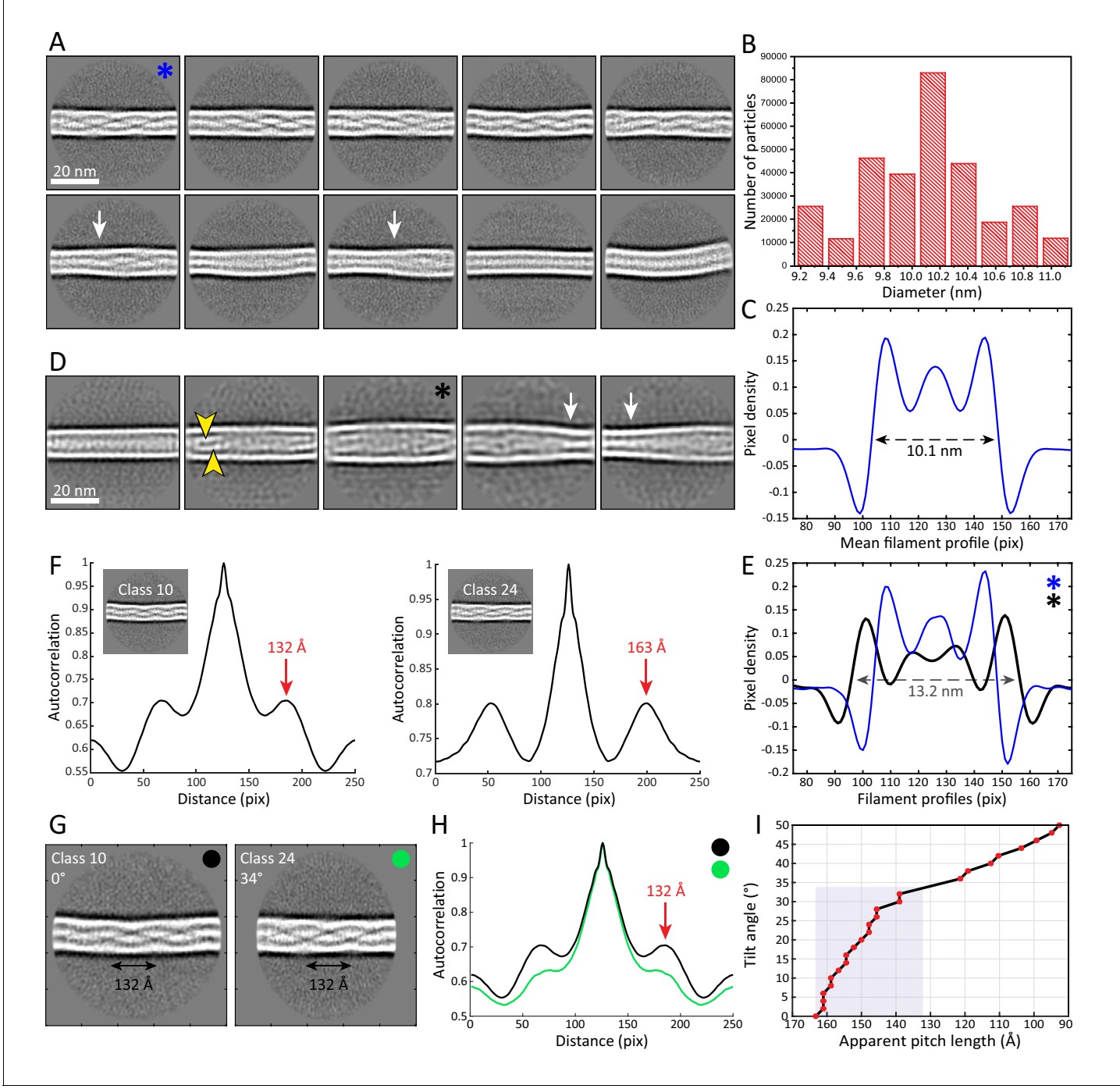

**Figure 2.** The architecture and heterogeneity of keratin filaments. (**A**) Ten of the most populated 2D class averages of keratin segments. High electron density is shown in white. Arrows indicate transition regions between helical and straight-line patterns. In total, 50 classes containing keratin segments (n=305,495) were collected (*Figure 2—figure supplement 1A*). (**B**) Distribution of filament diameters as measured in 50 2D class averages (*Figure 2—figure supplement 1A*). On the y-axis, the number of individual particles that constituted the 2D classes is plotted. (**C**) Mean intensity line-profile through all classes used in (**B**). The mean filament diameter (10.1 nm) is indicated and was measured between the zero-crossings of the curve. (**D**) Subset of keratin class averages showing larger filament diameters. Two individual filamentous densities are often detected within a filament (yellow arrowheads). Additionally, transition regions between thinner and thicker filament regions are detected (white arrows). (**E**) Intensity line-profiles through a narrow and a wide class indicated by blue and black asterisk (in (**A**) and (**D**)), respectively. Diameters of 10.1 nm and 13.2 nm (arrow) were detected. (**F**) Autocorrelation spectra of the displayed keratin classes (insets). Peaks of the autocorrelation function corresponding to the distance between repetitive elements along the filament (pitch) are indicated (arrows). (**G**) To show that out-of-plane tilting of KIFs can shift the autocorrelation peaks, Class 24 was tilted in silico by 34°, while Class 10 is untilted. The apparent pitch of both classes is indicated. (**H**) Autocorrelation spectra of the classes

*Figure 2 continued on next page*

*Figure 2 continued*

shown in (**G**). After tilting of Class 24 by 34°, both classes show an autocorrelation peak at the same marked position, an indicator that filament tilting might be the reason for the different pitches observed in the 2D classes. Green and black dots indicate which curve belongs to which class in (**G**). (**I**) Dependence of the apparent pitch length (autocorrelation peaks) on the filament tilt angle, measured by tilting Class 24 from 0° to 50° and calculating corresponding autocorrelation spectra. The gray area indicates the range of pitches found in keratin 2D classes.

The online version of this article includes the following figure supplement(s) for figure 2:

**Figure supplement 1.** 2D structural analysis of keratin segments.

pitch length would reflect the untilted filament, while all other repeating patterns would originate from different degrees of tilting. Based upon this reasoning, we tilted and projected the class with a pitch of 163 Å in silico and retrieved similar pitch lengths as seen in the real classes (*Figure 2I*). A tilt of up to ~34° can induce shortening of the pitch length from 163 Å to 132 Å (*Figure 2G,H*). Therefore, tilting between 0° - 34° would explain the variations that were detected in the repeating pattern of the keratin classes (*Figure 2I*). With ice thicknesses of up to ~300 nm and a minimal apparent persistence length of ~118 nm, this amount of tilting can be expected, and further analysis by cryo-ET revealed that even higher degrees of tilting are possible (see below). A helical pitch of ~163 Å agrees well with previous studies that suggested a pitch of ~162 Å for keratin filaments (*Aebi et al., 1983*; *Parry et al., 2001*).

## Computationally reconstituting keratin filaments from the class averages

The 2D class averages allowed us to identify structural differences in 55 nm long keratin segments. To understand how these structural features are organized at the level of long keratin filaments, it was important to determine how the class averages are arranged along keratin filaments which are up to several hundreds of nanometers in length. For this purpose, we utilized a back-mapping strategy that permits the computational reconstitution of the original filament out of 2D class averages (*Figure 2—figure supplement 1A,B*; *Kronenberg-Tenga et al., 2021*; *Martins et al., 2021*). Therefore, every segment was represented by its corresponding 2D class image, which was inversely transformed, so that it matched the original orientation of the raw segment. Then it was plotted at the original coordinate position, where the raw segment was selected from the electron micrographs. In this fashion, we assembled the original keratin filaments made out of the respective 2D class averages (*Figure 3—figure supplement 1A*), which were subsequently extracted and straightened. Since these computationally reconstituted filaments are assembled from class averages, their signal-to-noise ratio is drastically improved compared to the raw filaments. This approach allowed us to study long stretches of keratin filaments with improved resolution up to ~12 Å (*Figure 3*, *Figure 2—figure supplement 1C*).

The appearance of these computationally reconstituted keratin filaments is very heterogenous (*Figure 3*, *Figure 3—figure supplement 1*). Overall, they consist of patches of helical regions with clear repetitive patterns (*Figure 3A*, red, **E**), which are frequently interrupted by straight patterned stretches with less defined features (*Figure 3A*, yellow, **E**). The helical as well as the straight-line stretches are variable in length and frequency. While some filaments consist of mostly helical stretches, others are mixed or exhibit a mostly straight-line appearance (*Figure 3E*, *Figure 3—figure supplement 1B*). Additionally, the diameter fluctuates along a single filament (*Figure 3B,D*). For example, KIFs have regions of increased width up to 13.2 nm that often allow the identification of individual sub-chains (*Figure 3B,E*), as well as thinner regions with widths down to 9.2 nm (*Figure 3D,E*). The thinner regions usually display a straight pattern, whereas not all straight regions show a decrease in diameter.

Interestingly, the computationally reconstituted filaments revealed that helical regions with different pitch lengths, identified in the 2D class averages (*Figure 2*), can co-exist along a single filament (*Figure 3C*). This indicates that individual filaments changed their tilt angle along the course of the filament and ran through different z-heights of the ghost cell volume. Helical patterns with different pitches, indicating different tilt angles, lie in close proximity along KIFs, where they appear to transition smoothly into each other. These structural transitions reveal that keratin filaments constantly

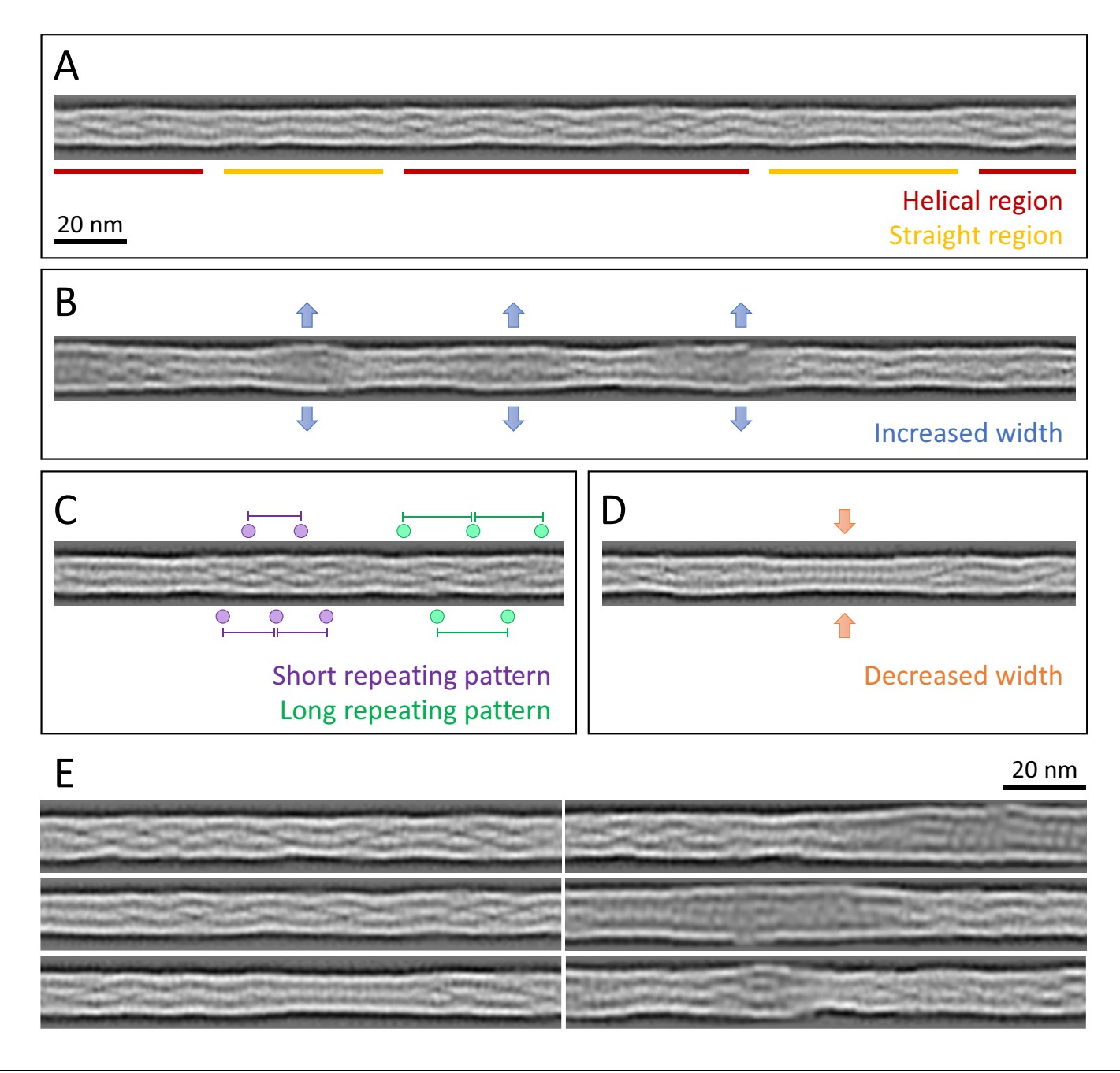

**Figure 3.** Structural polymorphism along keratin filaments. Computationally reconstituted filaments provide a realistic view of the KIFs at higher resolution (see Materials and methods). (A – D) Scale bar: 20 nm. (A) A typical keratin filament consisting of various regions with helical and straight-line patterns, indicated by red and yellow lines, respectively. (B) Keratin filament displaying diameter fluctuations. Areas of increased diameter are indicated (blue arrows). (C) Keratin filament showing helical patterns exhibiting different pitches, indicating modulations within the ice layer. Short repeating patterns (purple) indicate higher tilt angles in comparison to longer repeating patterns of in-plane filament stretches (green). (D) Keratin filament displaying diameter fluctuations. Areas of decreased diameter are indicated (orange arrows). (E) A collage of six computationally reconstituted keratin filaments showing structural diversity (total n=4460 filaments).

The online version of this article includes the following figure supplement(s) for figure 3:

**Figure supplement 1.** Reconstitution of keratin filaments.

fluctuate in the z-direction (perpendicular to the plane of the EM grid) and thus appear to be as flexible in the z-direction as they are in the *xy* plane (parallel to the EM grid).

Overall, computationally reconstituted keratin filaments reveal an enormous amount of structural heterogeneity (*Figure 3E*, *Figure 3—figure supplement 1C*). Every filament examined displays a unique phenotype, which demonstrates that keratin filaments are as versatile as the challenges they encounter in a living cell.

## Keratin filaments are cylindrical tubes with an internal electron dense core

A careful analysis of our datasets revealed several cryo-EM micrographs and multiple cryo-tomograms (21 out of 44) which contain KIFs that undergo a 90° turn along the thickness of the sample and therefore allow the observation of direct perpendicular cross sections of the filaments (*Video 2*). This behavior is quite remarkable, as it was never seen in tomograms (n = 225) of cellular vimentin intermediate filaments, imaged within detergent extracted mouse embryonic fibroblasts (MEF) (*Eibauer et al., 2021*, *Figure 4—figure supplement 1A,B*). Analysis of cross-sections from cryo-EM micrographs and cryo-tomograms revealed that keratin filaments are cylindrical tubes, in which an internal electron dense core is found (*Figure 4A–E*). This finding agrees with previous studies, which predicted that keratins contain internal mass, but less than anticipated for a completely filled filament (*Parry and Steinert, 1999*; *Engel et al., 1985*; *Bruce Fraser et al., 2003*; *Parry, 1996*; *Fraser and Parry, 2017*; *Herrmann and Aebi, 2004*; *Watts, 2002*) and was previously reported for keratin filaments of the living layers of human epidermis and the stratum corneum (*Norlén and Al-Amoudi, 2004*). Moreover, we could identify individual sub-filaments, which form a hexameric ring structure in cross-section (*Figure 4B*). Based on this geometry and previous literature, it is likely that the sub-structures represent tetrameric protofilaments and therefore the mature filament would be composed of ~six protofilaments to yield ~24 polypeptides in cross-section (*Parry and Steinert, 1999*). This agrees with previous mass-per-unit-length analyses of epidermal keratins and keratins from simple epithelia (*Herrmann et al., 1999*; *Engel et al., 1985*). To study this phenomenon in more detail, we analyzed the cross-sectional views detected in cryo-tomograms. We found that depending on the individual filament and tomographic slice, the number of visible protofilaments varies, which might indicate polymorphism, that is a variable number of protofilaments building the KIF. However, it might also be an imaging artifact originating from the low signal-to-noise ratio of individual sections, that aggravates the identification of the thin protofilaments. Nevertheless, the tomographic slices yielded some unique insights into filament arrangement. When following a filament in cross section through the tomographic volume, in certain slices densities of neighboring protofilaments seem to merge into one continuous structure, indicating that there are positions along the filament where the protofilaments are interacting so tightly that they could not be resolved individually (*Figure 4E*, *Figure 4—figure supplement 1C*). Other positions reveal more than six protofilaments, which may reflect overlap regions between tetramers along the keratin filaments, yielding the impression of more strains, or filament regions containing additional protofilaments. To study the position of the central density within the keratin tube, we extracted a cross-sectional filament (labeled with E in *Figure 4D*) and rotated it precisely into 90° cross-section. Projection of 4.2-nm-thick slices along the filament revealed that the central density is not always found in the exact center of the filament, but is sometimes detected closer to the filament walls (*Figure 4—figure supplement 1E,F*). This indicates, that the central density moves within the keratin tube and is not fixed to the center of the filament.

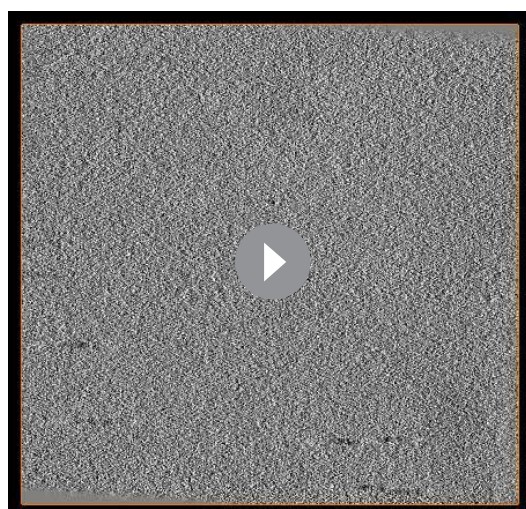

**Video 2.** Cryo-tomogram showing the modulations of the keratin filaments within the ice layer.
https://elifesciences.org/articles/70307#video2

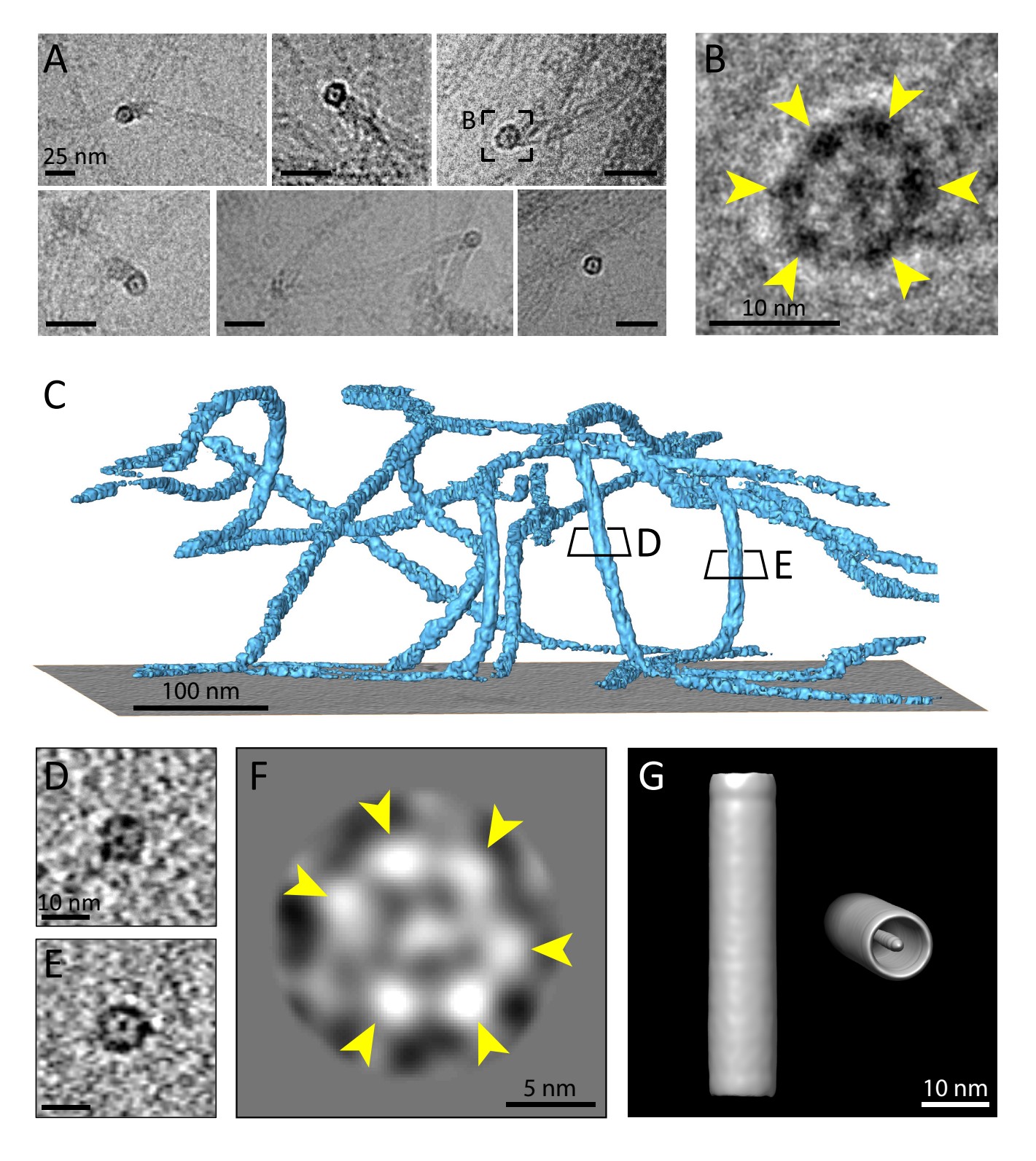

**Figure 4.** Multiple protofilaments and an internal electron dense core are canonical components of keratin filaments. (**A**) Cross-section views of keratin filaments detected within the cryo-EM micrographs of ghost cells. An electron dense core is visible in the center of the keratin tube. Scale bars: 25 nm. (**B**) Zoomed-in view of the area boxed in (**A**). The cross section view reveals an internal core surrounded by six protofilaments as constituents of the tube (yellow arrowheads). (**C**) A surface rendered tomogram of a ghost cell was rotated in order to show the modulation of the keratin filaments within the

*Figure 4 continued on next page*

*Figure 4 continued*

ice layer (n=44). The three-dimensional keratin network is visualized (light blue). The level of the support is shown as a gray colored slice. Tomographic slices through vertically oriented filaments showing cross section views are indicated by boxes. (D) - (E) 7 nm thick xy-slices of the areas indicated in (C), showing KIFs as tube-like structures with a central density. Individual protofilaments can be identified. Scale bars: 10 nm. (F) A 2D class average of cross-section views extracted from individual regions of vertically oriented filaments (n=19), revealing the six individual protofilaments constituting the keratin filament tube (yellow arrowheads). (G) Low-resolution 3D model indicating the overall dimensions of a keratin filament and the presence of the central density. The structure was calculated template-free by randomizing the rotation angle of extracted 55 nm long keratin segments. Left: Side view. Right: Tilted cross section view revealing internal electron dense core.

The online version of this article includes the following figure supplement(s) for figure 4:

**Figure supplement 1.** Keratin filaments show a unique flexibility and contain a central density.

To study the number of protofilaments in more detail, we selected 710 cross-sectional views of filaments found within the cryo-tomograms, extracted central slices and subjected them to 2D classification, revealing a symmetric hexameric class (*Figure 4F*). Six individual protofilaments can be clearly distinguished in the ring (arrowheads), which agrees with our studies of the raw data. As expected, additional classes were found showing deviations from this hexameric structure (*Figure 4—figure supplement 1D*). These structural differences may represent actual changes in symmetry and a polymorphic composition of KIFs, but also deviations from a perfect perpendicular cross-section may account for this observation.

Finally, to get an impression of a mature keratin filament in three dimensions, we generated a low-resolution 3D model of a keratin filament from our single particle dataset of 55 nm long segments using the Relion software package (*Scheres, 2012*; *He and Scheres, 2017*). By randomizing the rotational angle along the filament axis, a template-free unbiased model was generated (*Figure 4G*). The 3D model strengthens our findings that keratin filaments are formed as hollow cylinders with a central electron density. Due to randomization of the rotation angle, the individual protofilaments are not resolved in this structure, however, it provides a view of the central density and the overall dimensions of the K5/14 intermediate filaments.

## Discussion

Keratin intermediate filaments are major components of the cytoskeleton that are involved in many cellular processes (*Redmond and Coulombe, 2021*; *Gordon, 2020*; *Danielsson et al., 2018*). However, due to their flexibility, heterogeneity and yet to be resolved symmetry, a high-resolution structure of keratin filaments in their native state has not been obtained to date. In this study, we describe novel insights into the architecture of in vivo assembled K5/K14 filaments by imaging them directly within ghost cells. This approach has enabled us to study native KIFs containing all their post-translational modifications, which are known to play an important role in their assembly and function (*Snider and Omary, 2014*). In addition, this preparation circumvents the need for denaturing and renaturing the keratin proteins prior to in vitro assembly, a step which likely increases structural polymorphism (*Parry and Steinert, 1999*). Since the preparation of ghost cells involves a short detergent exposure of 15–20 s, we showed that the well-established structure of F-actin is retained and can be resolved to 6.1 Å. Similarly, the structure of lamin filaments is also preserved using this approach (*Kronenberg-Tenga et al., 2021*; *Turgay and Medalia, 2017*). Therefore, we concluded that the structure of keratin IFs is unlikely to be affected by the sample preparation procedure.

The K5/K14 IFs filaments are known to assemble in the cell periphery (*Windoffer et al., 2004*) to form bundles, networks and single IFs in all cellular domains (*Ma et al., 2001*; *Yamada et al., 2002*; *Lee and Coulombe, 2009*). In this study, we have focused on individual KIFs located both in peripheral and more central regions as KIF bundles are dense, highly complex structures and would be unsuited for our averaging procedures (*Sigworth, 2016*). Individual filaments were found to be very flexible, which agrees with previous studies (*Ma et al., 2001*; *Coulombe and Fuchs, 1990*; *Herrmann et al., 2002*; *Köster et al., 2015*), and showed a high degree of bending within a few hundred nanometers. We determined the minimal apparent persistence length of highly bent KIFs to be 118.4 ± 39.2 nm, indicating that intact keratin filaments can undergo a 90° turn within a distance of 2–3 dimer lengths (dimer length ~44 nm *Bray et al., 2015*; *Quinlan et al., 1984*). This value is slightly lower than the persistence length determined for K8/K18 filaments (300–650 nm)

(*Block et al., 2015*; *Lichtenstern et al., 2012*; *Pawelzyk et al., 2014*), which can be attributed to the fact that the K5/K14 filaments were still embedded into the cellular filament network.

The high flexibility and modulation of keratin filament orientation is also apparent as filaments can span through the entire thickness of the ice on a cryo-EM grid. The filaments often do not lie flat on the grid, but are frequently tilted out of the *xy* plane and form a wavy network in all three dimensions. Cryo-tomograms of ghost cells allowed us to follow individual filaments through different heights of the cell and show that the filaments can undergo 90° turns within a thickness of <300 nm (*Figure 4*). Interestingly, other IFs such as vimentin seem to be fluctuating less through the different heights of the ghost cells (*Figure 4—figure supplement 1A,B*).

The bending ability of keratin filaments has enabled us to analyze perpendicular cross-section views and therefore directly reveal that they are built from six sub-filaments surrounding an electron dense core. A hexameric filament arrangement accompanied by a central core has been previously observed in keratin filaments of the human stratum corneum (*Norlén and Al-Amoudi, 2004*). This supports a structure comprised of six tetrameric protofilaments, yielding 24 polypeptides in cross section. In support of this finding, mass-per-unit-length studies identified 21–25 polypeptides per cross-section of reassembled epidermal keratin extracts and in recombinant K8/K18 IFs prepared in vitro (*Herrmann et al., 1999*; *Engel et al., 1985*). It is unlikely that the identified sub-filaments represent protofibrils, that is octameric assemblies, as this would yield 48 polypeptides in cross section, which does not agree with the MPL studies. The proposed model of a 'universal IF', consisting of 32 polypeptides in cross section, is not supported by our data. While this model agrees well with previous studies of other IFs, for example vimentin (*Herrmann et al., 1999*; *Goldie et al., 2007*; *Steven et al., 1982*; *Kooijman et al., 1997*), it does not fit with studies of keratin filaments. Therefore, we conclude that different members of the IF superfamily show substantial diversity with respect to their structure (*Goldie et al., 2007*). Although MPL studies reveal that a small number of keratin filaments may contain ~32 polypeptides in cross-section, both MPL and cryo-EM studies including our own show that the majority of KIFs contain ~24 polypeptides in cross-section, organized into ~six protofilaments (*Herrmann et al., 1999*; *Engel et al., 1985*; *Norlén and Al-Amoudi, 2004*). Early studies of Aebi et al proposed a model of KIFs consisting of 8 protofilaments, arranged in four protofibrils, based on EM studies of unraveled filaments (*Aebi et al., 1983*).

It is noteworthy that keratin filaments are not completely filled, but possess a distinct density in the center that is separated from the protofilaments forming the filament tube A central core in KIF cross-sections has also been reported in the viable cell layers and stratum corneum of human epidermis (*Norlén and Al-Amoudi, 2004*). As the viable cell layers of the epidermis express mostly K5/K14 filaments, this lends further support that the central core is a canonical structure of K5/K14 IFs. An internal core has also been identified in trichocyte keratins found in wool (*Bruce Fraser et al., 2003*; *Rogers, 1964*; *Kadir et al., 2017*). Previous cryo-ET studies of in vitro assembled vimentin filaments indicated no central core subunit (*Goldie et al., 2007*), however, a recent cryo-ET analysis of cellular assembled vimentin filaments also identified a frequently occurring central density (*Eibauer et al., 2021*). The central density described in the present study may correspond to an additional tetrameric protofilament (*Bruce Fraser et al., 2003*) or another cellular component. Interestingly, this density is not fixed to the exact center of the keratin tube, but small changes in its position were identified along the keratin filaments.

The 2D class averaging of keratin segments revealed structural variations that can be divided into multiple unique phenotypes. This is a remarkable property, that becomes even more apparent when comparing keratins to other cytoskeletal elements: While all analyzed actin class averages highly resemble each other, for example, showing the same helical pattern, filament organization and diameter, each keratin class average seems to be unique and to reflect a different structural arrangement of keratin filaments. The most pronounced feature is the helical repeating pattern in some 2D classes, while other classes show no apparent helical symmetry. Computational reconstitution of long keratin filaments revealed that these regions alternate along the same filaments, indicating that both patterns are crucial components of the keratin structure. This finding demonstrates that the heterogeneity and potential polymorphism does not lie between filaments, but within filaments. 2D classification and computational filament reconstitution additionally revealed large heterogeneities in the filament thickness. Although the most prevalent diameter detected in keratin filaments is 10.1 nm, the diameter can fluctuate between 9.2 and 13.2 nm. This type of heterogeneity has been previously described for several types of IFs (*Herrmann et al., 1999*; *Engel et al., 1985*; *Steven et al.,*

*1983*) and is thought to reflect a varying number of subunits per cross-section. However, our results show that regions of increased diameter frequently yield insights into the subunit organization of the filament, indicating that the individual protofilaments are more loosely packed. The increased widths may therefore also reflect regions that are in a state of assembly or subunit exchange (*Çolakoğlu and Brown, 2009*; *Ngai et al., 1990*; *Miller et al., 1991*; *Vikstrom et al., 1989*). Further, they might reflect areas of local post-translational modifications or regions where filaments were locally damaged. Interestingly, these regions are not restricted to the edges of filaments, but can occur in the mid regions of already assembled filaments.

Filament stretches with diameters smaller than 10 nm might reflect either supercoiling or regions that have experienced a greater degree of localized mechanical stress. Previous studies showed that upon stretching beyond 200%, keratins adopt a plastic behavior that is accompanied by strain hardening and a significant reduction in diameter (*Block et al., 2015*; *Kreplak et al., 2005*; *Fudge et al., 2003*), which is thought to be mainly a result of α-helix to β-sheet transitions of the coiled-coil domains (*Kreplak et al., 2005*; *Fudge et al., 2003*; *Pinto et al., 2014*). Our experiments were conducted in ghost cells after forces were relaxed, therefore, only irreversible unfolding can be detected. Thus, thinner regions in keratin filaments may reflect local domains where extensive forces were applied to the filament network and the filaments adapted by irreversibly unfolding their coiled-coil domains, leading to a reduction in diameter.

Overall, our results demonstrate that structural polymorphism is an intrinsic property of keratin filaments that were assembled within epithelial cells, and not only occurs in filaments that were assembled in vitro. When compared to the other components of the cytoskeleton, microtubules and actin filaments, which adapt a highly uniform arrangement, the structural heterogeneity of keratin filaments is clearly exceptional. Structural flexibility and changes in helical packing of the filament may provide structural support for the elastic nature of the keratin network and would help to explain their high resistance to breakage (*Etienne-Manneville, 2018*; *Block et al., 2015*). This feature of keratin filaments would coincide with their task to adapt to different mechanical stresses while maintaining a stable network.

Our findings demonstrate the importance of determining a high-resolution structure of keratin filaments in order to understand the details of their assembly states and the functional significance of their heterogeneity in cells. Analyzing in vivo assembled filaments provides an approach to study the structure of keratins in their native state with their original post-translational modifications. Understanding the high-resolution structure of keratin filaments would also provide a foundation for determining how keratin mutations affect their structure and how they interact with binding partners. However, with reference to the findings of the current study, we want to emphasize that it is likely that there is not a single 3D structure that can describe keratin filaments in their entirety. In contrast, it is more likely that the conformational range needs to be described by a multitude of structures. Cryo-EM and cryo-ET are the methods of choice for unravelling the complexities of the 3D structure of mature keratin filaments, as the coordinated use of these techniques can resolve both their flexibility and heterogeneity. Given the rapid advances in cryo-EM imaging, sample preparation and image processing, we anticipate that the structural analysis of keratin intermediate filaments will continue to provide new insights into their cellular structure and functions.

# Materials and methods

**Key resources table**

| Reagent type (species) or resource | Designation | Source or reference | Identifiers | Additional information |
|---|---|---|---|---|
| Gene (*Mus musculus*) | *Krt6a* | | UniProtKB - P50446 | Targeted by CRISPR/ Cas9 Knockout |
| Gene (*Mus musculus*) | *Krt6b* | | UniProtKB - Q9Z331 | Targeted by CRISPR/ Cas9 Knockout |
| Strain, strain background (*E. coli*) | DH5α | ThermoFisher Scientific | Cat# 18265017 | Chemically competent cells |

*Continued on next page*

*Continued*

| Reagent type (species) or resource | Designation | Source or reference | Identifiers | Additional information |
|---|---|---|---|---|
| Cell line (*Mus musculus*) | KtyI KO K14 | *Homberg et al., 2015*. DOI: 10.1038/jid.2015.184 | | T. Magin lab, Institute of Biology, University of Leipzig, Germany. Cell line used to generate K5/K14_1 cell line |
| Cell line (*Mus musculus*) | K5/K14_1 | This paper | | Clonal knockout cell line of *Krt6a* and *Krt6b*, maintained in the Medalia lab |
| Antibody | Anti-mouse Keratin 14 (mouse monoclonal) | ThermoFisher Scientific | MA5-11599, Clone LL002 | (1:100 – 1:10) |
| Antibody | Anti-mouse Keratin 5 (rabbit polyclonal) | BioLegend | Cat# 905503 | (1:500) |
| Antibody | Anti-mouse Keratin 6a (rabbit polyclonal) | BioLegend | Cat# 905702 | (1:500) |
| Antibody | Cy3 AffiniPure anti-rabbit (donkey polyclonal) | Jackson Immuno Research | Cat# 711-165-152 | (1:100) |
| Antibody | FITC AffiniPure anti-mouse (donkey polyclonal) | Jackson Immuno Research | Cat# 715-095-150 | (1:100) |
| Recombinant DNA reagent | pX458 (pSpCas9(BB)—2A-GFP) (plasmid) | Addgene | Cat# 48138 | CRISPR/Cas9 knockout |
| Recombinant DNA reagent | pGEM-T Easy (plasmid) | Promega | Cat# A1360 | |
| Sequence-based reagent | guideRNA insert targetting *Krt6a* and *Krt6b* gene | This paper | | guideRNA sequence: GAGCCACCGCTG CCCCGGGAG. guideRNA cloned into pX458 plasmid and transfected into KtyI KO K14 cells. |
| Commercial assay or kit | P3 primary cell 4D-Nucleofector kit | Lonza | Cat# V4XP-3032 | Transfection of KtyI KO K14 cells |
| Commercial assay or kit | jetPRIME | Polyplus transfection | Cat# 114–07 | Transfection of KtyI KO K14 cells |
| Commercial assay or kit | GenElute Mammalian Genomic DNA kit | Sigma-Aldrich | Cat# G1N70-1KT | Genomic DNA extraction |
| Chemical compound, drug | 99.9% anhydrous methanol | Alfa Aesar | Cat# 41838 | Fixation of mammalian cells |
| Chemical compound, drug | Hoechst 33342 | Sigma-Aldrich | Cat# B2261 | (1:10000) |
| Chemical compound, drug | Dako mounting medium | Agilent | Cat# S3023 | |
| Chemical compound, drug | Prolong Glass Anti-Fade | ThermoFisher Scientific | Cat# P36980 | |

*Continued*

| Reagent type (species) or resource | Designation | Source or reference | Identifiers | Additional information |
|---|---|---|---|---|
| Software, algorithm | TIDE webtool | Netherlands Cancer Institute | https://tide.nki.nl/ | CRISPR/Cas9 knockout analysis |
| Software, algorithm | FIJI | Max Planck Institute of Molecular Cell Biology and Genetics, Dresden, Germany | RRID:SCR_002285 | For data analysis |
| Software, algorithm | Illustrator | Adobe Inc | RRID:SCR_010279 | For data analysis |
| Software, algorithm | OriginPro | OriginLab | RRID:SCR_014212 | For data analysis |
| Software, algorithm | MATLAB | MathWorks | RRID:SCR_001622 | For data analysis |
| Software, algorithm | RELION | *Scheres, 2012.* DOI: 10.1016/j.jsb.2012.09.006 | RRID:SCR_016274 | For data analysis |
| Software, algorithm | IMOD | University of Colorado Boulder, Colorado, USA | RRID:SCR_003297 | For data analysis |
| Software, algorithm | Amira | ThermoFisher Scientific | RRID:SCR_007353 | For data analysis |
| Software, algorithm | SerialEM | University of Colorado Boulder, Colorado, USA | RRID:SCR_017293 | For data acquisition |
| Software, algorithm | UCSF Chimera | University of California, California, USA | RRID:SCR_004097 | For structure visualization |

## Generation of the K5/K14_1 cell line

Mouse keratinocytes lacking the entire type I keratin cluster (KtyI KO) but stably transfected with K14 (*Homberg et al., 2015*) were authenticated using transcriptome profiling and checked for mycoplasma contamination in the T. Magin lab (Institute of Biology, University of Leipzig, Germany). KtyI KO K14 cells were cultured on Collagen I (bovine, CellSystems) coated dishes at 32°C and 5% $CO_2$ in calcium-depleted FAD medium in the presence of puromycin (LabForce, 8 µg/ml medium). Confluent cells were trypsinized using 2.5 x trypsin/EDTA solution (Sigma-Aldrich, T4174) and re-seeded at a maximum splitting ratio of 1:2. To knock-out the *Krt6a* and *Krt6b* genes, cells were transfected with the pX458 (pSpCas9(BB)−2A-GFP) plasmid (Addgene) carrying a GFP-tagged Cas9 and a guideRNA insert targeting both the *Krt6a* and *Krt6b* gene (guideRNA sequence: GAGC-CACCGCTGCCCCGGGAG). Transfection using electroporation was performed according to the manufacturer's protocol using a P3 primary cell 4D-Nucleofector kit (Lonza) and program 138 for human keratinocytes, followed by another round of transfection using jetPRIME (Polyplus transfection). Next, genomic DNA was extracted from clonal cell lines using the GenElute Mammalian Genomic DNA kit (Sigma-Aldrich) and the *Krt6a* and *Krt6b* gene fractions where the indel mutations were expected were amplified by PCR. PCR fragments were sequenced (Microsynth) and the indel mutation spectrum was analyzed using the TIDE webtool (https://tide.nki.nl/). The K5/K14_1 clone was identified as homogenous K6a knockout and heterogenous K6b knockout and was therefore used for all studies. To further verify the knockout, PCR fragments were cloned into the pGEM-T Easy vector (Promega, A1360) and amplified in DH5α cells. Bacterial clones carrying individual gene sequences of *Krt6a* or *Krt6b* were picked and amplified, plasmids were extracted and sequenced (Microsynth). By analyzing 19 *Krt6a* and 22 *Krt6b* sequences, the homogenous K6a and heterogenous K6b knockout were verified.

## Immunostaining

The KtyI KO K14 and K5/K14_1 cells were seeded on Collagen I coated glass cover slips in cell culture dishes and incubated overnight at 32°C and in 5% $CO_2$. For staining with keratin antibodies, cells were fixed for 5 min using ice-cold 99.9% anhydrous methanol (Alfa Aesar, 41838). Non-specific antibody binding sites were blocked by incubating the cover slips for 30 min in blocking buffer (1% BSA, 22.52 mg/ml glycine in PBS with 0.1% Tween (PBS-T)). Next, cover slips were incubated for 1 hr at room temperature with mouse anti-mouse Keratin 14 (LL002, Thermo Fisher, MA5-11599, 1:100 – 1:10), rabbit anti-mouse Keratin 5 (BioLegend, 905503, 1:500) or rabbit anti-mouse Keratin 6a (BioLegend, 905702, 1:500) in 1% BSA in 0.1% PBS-T. It should be noted here that the K6a antibody used is a polyclonal antibody, which is suspected to bind to the K6b protein as well, due to their high sequence identity (92.6%). After 3x 5 min washing steps in PBS, cover slips were incubated with Cy3 donkey anti-rabbit (Jackson Immuno Research, 711-165-152, 1:100) or FITC donkey anti-mouse (Jackson Immuno Research, 715-095-150, 1:100) secondary antibodies in 1% BSA in 0.1% PBS-T. Cells were washed 3x for 5 min in PBS, before nuclei were stained with Hoechst 33342 (Sigma-Aldrich, B2261, 1:10,000) for 10–20 min at room temperature. After a final wash step 3x for 5 min in PBS, the cover slips were mounted on glass slides with Dako mounting medium (Agilent, S3023) or Prolong Glass Anti-Fade (Thermo Fisher, P36980). Confocal imaging for *Figure 1* was carried out with a laser scanning confocal microscope (Nikon A1R confocal microscope, Nikon) using an oil immersion objective lens (Plan Apo 60X Oil objective, 1.4 NA, Nikon). Keratin was excited with a 488 nm wavelength laser and the optical sections were imaged at 100 nm intervals. Maximum intensity projections of the Z-stacks are presented. Keratin networks for *Figure 1—figure supplement 1C* were imaged using a spinning disk confocal laser scanning microscope (Olympus IXplore SpinSR10 with YOKOGAWA CSU-W1 spinning disk). 3D confocal stacks were acquired with a UPLSAPO UPlan S Apo 60x/1.3 OIL objective (Olympus). Fluorescent proteins were excited at 405 nm (50 mW, 10% laser power), 488 nm (100 mW, 15% laser power) and 561 nm (100 mW, 5% laser power).

## Sample preparation for cryo-EM and cryo-ET

K5/K14_1 cells were seeded on glow-discharged Collagen I coated holey carbon gold EM grids (Au R2/1, 200 mesh, Quantifoil) and incubated overnight at 32°C and in 5% $CO_2$. The grids were rinsed in washing buffer (1x PBS, 2 mM $MgCl_2$), cells were permeabilized for 15–20 s in permeabilization buffer (1x PBS, 0.1% Triton X-100, 600 mM KCl, 10 mM $MgCl_2$ and protease inhibitors), and rinsed again in washing buffer. Next, the grids were incubated with 2.5 units/µl benzonase (Merck, 71206–3) in washing buffer for 30 min and washed in PBS before vitrification in liquid ethane using a manual plunge freezing device. For cryo-ET samples, 10 nm gold fiducial markers (Aurion, Netherlands) were added to the grids right before freezing.

## Cryo-EM and cryo-ET data acquisition

The grids were analyzed using a 300 kV Titan Krios electron microscope (Thermo Fisher) equipped with a K2 Summit direct electron detector (Gatan) mounted on a post-column energy filter (Gatan). Cryo-EM micrographs were acquired in zero-loss energy mode using a 20 eV slit. Data were recorded with SerialEM 3.5.8 in low-dose mode (*Mastronarde, 2005*). Overview images were acquired, in which the filaments were identified prior to data acquisition. Micrographs were acquired at nominal magnifications of ×46,511 with a pixel size of 1.075 Å, ×28,571 with a pixel size of 1.75 Å, and ×22,665 with a pixel size of 2.206 Å. A defocus range between −0.5 and −3.5 µm was chosen. Dose-fractionation was used with a frame exposure of 0.2 s with a total exposure time of 10 s (50 frames in total). This corresponds to a total electron dose of ~20 e/Å$^2$ for the ×22,665 dataset, ~41 e/Å$^2$ for the ×28,571 and ~84 e/Å$^2$ for the ×46,511 dataset. In total, data were collected from ~60 ghost cells. Micrographs acquired at a magnification of 22,665 x were used for most experiments conducted in this study, as stated below. The datasets acquired at ×46,551 and ×28,571 magnification were used to visualize the keratin cytoskeleton in closer detail and micrographs from these datasets are shown in *Figure 1*, *Figure 1—figure supplement 1* and *Figure 4*.

Tilt series were acquired in zero-loss energy mode with a 20 eV slit at a nominal magnification of ×28,571, resulting in a pixel size of 1.75 Å and a defocus of −3 µm. A bidirectional tilt scheme with a tilt range of ± 60° and an increment of 3° was chosen, corresponding to 41 projections per tilt

series and a total accumulative electron dose of ~89 e/Å$^2$. SerialEM 3.5.8 in low dose mode was used for data acquisition (**Mastronarde, 2005**).

## Minimal apparent persistence length measurements

Highly bent keratin filaments were identified in electron micrographs and traced with Fiji using the segmented line tool including spline fit (**Schindelin et al., 2012**). Minimal filament contour lengths that undergo a 90° turn were traced. The persistence length is defined as the distance along a filament at which the tangent-tangent correlation function along the contour length decays, this occurs after a 90° turn (**Reisner et al., 2012**). However, since our sample is out of equilibrium, as individual filaments are entangled in a network and absorbed to the EM grid, and filaments are imaged in 2D, only an apparent persistence length is described. Further, only highly bent filaments were considered in this analysis, yielding a minimal apparent persistence length, as the whole filament population is diverse and cannot be described as a single state.

## Cryo-EM data processing

1860 cryo-EM micrographs acquired from ~20 cells, at a magnification of ×22,665 were processed with RELION 2.1 and RELION 3.0 using the helical toolbox (**Scheres, 2012**; **He and Scheres, 2017**; **Zivanov et al., 2018**). At the initial stages of analysis, individual micrographs were excluded when no individual keratin filaments could be identified. Frame-based motion correction and dose-weighting were performed using MotionCor2 (**Zheng et al., 2017**). The contrast transfer function was estimated using CTFFIND4 (**Rohou and Grigorieff, 2015**). Low-quality micrographs showing high defocus, high astigmatism or low resolution (assessed by CTF estimation, information below 7.5 Å) were excluded, resulting in 1763 high-quality micrographs used for further processing steps. Keratin filaments were either picked manually or automatically using the RELION helical toolbox. To generate a template for autopicking, 55,073 keratin particles were picked manually as start-to-end helices, extracted with a box size of 250 pixels (~55 nm) and 2D classified twice to create classes with straight keratin segments. These classes served as a reference for automated picking of 505,211 particles. For manual picking, 298,056 particles were selected as start-to-end helices. Particles were extracted in boxes of 250 pixels, corresponding to ~55 nm, or 164 pix, corresponding to ~36 nm, with an inter-box distance of 50 Å. Iterative 2D classification procedures were performed, using a spherical mask of 500 Å or 356 Å, respectively.

Keratin filament segments, 55 nm in length, were classified to yield 305,495 particles in straight classes. Autocorrelation spectra were calculated with MATLAB (2019a, MathWorks). The filament diameter was measured by plotting vertical intensity line-profiles of all classes using MATLAB and measuring the area where the intensity lies above zero.

Intensity line-profiles of each class were generated by averaging all lateral sections through the segment. OriginPro 2018 software (OriginLab Corporation) was used to plot the diameter distribution. Instead of plotting the number of classes, which was set to a fixed value, the number of particles in each class was plotted, as it reflects how many particles with a certain diameter are present in the dataset. A mean intensity line-profile for *Figure 2C* was generated by averaging all classes from *Figure 2—figure supplement 1A*.

Segments with a box size of 36 nm were used for computational filament reconstitution. Filament reconstitution was performed as previously reported (**Kronenberg-Tenga et al., 2021**) and described below with classes of automatically, as well as manually, picked particles.

Actin filaments were processed identically to keratin filaments to guarantee comparability. To generate a template for autopicking, 22,228 actin particles were picked manually as start-to-end helices, extracted with a box size of 164 pixels (~36 nm) and 2D classified twice to create classes with straight actin segments. These classes served as a reference for automated picking of 693,903 particles. Particles were extracted in boxes of 164 pixels, corresponding to ~36 nm, with an inter-box distance of 50 Å. Multiple rounds of 2D classification were performed, using a spherical mask of 356 Å. A total of 405,044 particles from the highest resolved 2D classes were used for 3D classification into five classes. The highest resolved 3D class, containing 174,954 particles, was subjected to 3D refinement.

The final unmasked map showed a resolution of 7.38 Å, based on the gold standard Fourier shell correlation (FSC) 0.143 criterion (*Rosenthal and Henderson, 2003*; *Scheres, 2012*). The structure was sharpened to 6.13 Å using an isotropic B-factor of $-276$ Å$^2$.

## Cryo-ET data processing

Tilt series were processed using the IMOD workflow, including contrast transfer function (CTF) correction (*Kremer et al., 1996*). For visualization purposes, a SIRT-like filter according to 10 iterations was applied during tomogram reconstruction. Cellular structures present in the tomograms were manually segmented and visualized using the Amira 5.6.0 software package (Thermo Fisher Scientific). Tomogram movies were created using Amira 5.6.0 and Fiji (*Schindelin et al., 2012*). 710 cross section views of keratin filaments were picked in 21 tomograms and reconstructed as sub-tomograms using IMOD. Central 2D slices were extracted from the sub-tomograms and utilized for 2D classification in RELION. To assess the position of the central density within the keratin filament for *Figure 4—figure supplement 1E,F*, the filament labeled with E in *Figure 4D* was extracted in silico and rotated into precise 90° cross section. Afterwards, 4.2 nm thick slices were projected along the length of the filament. The resulting cross-sectional views were filtered with a gaussian blur using Fiji and analyzed for the position of the central density within the filament tube. Positions were represented by circles using Adobe Illustrator (Adobe Inc).

## Computational reconstitution of keratin filaments

To generate computationally reconstituted filaments a back-mapping strategy was pursued in MATLAB, using the ~36 nm long keratin segments which were used for 2D classification. First, all particles belonging to the same filament were grouped. Filament assignments were made based on the helical tube ID defined by RELION for every particle. Next, all particles belonging to the same filament were sorted in ascending order based on their picking coordinates. Then, their corresponding 2D class images were inversely transformed, so that their orientation matches the original orientation of the raw segments in the cryo-EM micrographs. Next, the 2D class images were plotted at the original coordinates of the particles. To remove background noise, the classes were masked in the y-direction and only the central 132 Å were plotted. Since particles were picked with inter-box distances of 50 Å, while 2D classes have a box size of 360 Å, neighboring segments would strongly overlap. To avoid this, classes were cropped to not extend into neighboring particle positions, and only a small amount of overlap of <four classes with soft edges was allowed to avoid cropped edges in slightly bent filaments. Reconstituted filaments were normalized to equal intensity. Next, a straightening procedure was applied as previously described to extract, align and straighten the reconstituted filaments (*Steinert et al., 1985*; *Kocsis et al., 1991*). Validity of this approach was ensured by using high-resolution actin classes as a control.

## Analyzing helical and straight-line patterns in individual filaments

Plots representing the order of helical and straight segments along individual filaments, represented by colored circles, were generated in MATLAB as previously described (*Martins et al., 2021*). 2D classes were grouped into helical or straight clusters based on their appearance. Next, each particle within a filament was represented by red or blue circles, depending on whether its corresponding 2D class belonged to the helical or straight cluster. In the analysis seen in *Figure 3—figure supplement 1B*, segments that originate from the same filament are plotted as columns of circles. Segments are sorted in ascending order based on their coordinates along the filament.

## 3D reconstruction of a keratin filament

305,495 uniform keratin segments from 55 nm boxes were selected by 2D classification (*Figure 2—figure supplement 1A*) and used for 3D reconstruction. To generate a low-resolution 3D filament model, the rotation angle along the filament axis of all particles was randomized to prevent preferred orientations. Next, a filament was reconstructed using relion_reconstruct. The 3D model was visualized using Chimera (*Pettersen et al., 2004*).

## Acknowledgements

The authors thank the Center for Microscopy and Image Analysis at the University of Zurich for providing support and equipment. Funding This research was funded by the Swiss National Science Foundation Grant (31003A_179418). MSW was supported by the Forschungskredit of the University of Zurich (FK-18–041). The Goldman laboratory is supported by grants 5PO1 GM096971 and RO1GM140108 from the National Institutes of Health.

## Additional information

### Funding

| Funder | Grant reference number | Author |
|---|---|---|
| Schweizerischer Nationalfonds zur Förderung der Wissenschaftlichen Forschung | 31003A_179418 | Ohad Medalia |
| NIH Office of the Director | 5PO1 GM096971 | Robert D Goldman |
| NIH Office of the Director | RO1GM140108 | Robert D Goldman |

The funders had no role in study design, data collection and interpretation, or the decision to submit the work for publication.

### Author contributions

Miriam S Weber, Data curation, Formal analysis, Validation, Investigation, Visualization, Methodology, Writing - original draft, Writing - review and editing; Matthias Eibauer, Data curation, Methodology, Writing - review and editing; Suganya Sivagurunathan, Data curation; Thomas M Magin, Resources, Writing - review and editing; Robert D Goldman, Ohad Medalia, Conceptualization, Funding acquisition, Project administration, Writing - review and editing

### Author ORCIDs

Miriam S Weber (iD) https://orcid.org/0000-0003-2125-8222
Ohad Medalia (iD) https://orcid.org/0000-0003-0994-2937

### Decision letter and Author response

Decision letter https://doi.org/10.7554/eLife.70307.sa1
Author response https://doi.org/10.7554/eLife.70307.sa2

## Additional files

### Supplementary files

• Transparent reporting form

### Data availability

Representative cryo-ET data have been deposited in the Electron Microscopy Data Bank under accession codes EMD-12958 and EMD-12959. In addition, data was uploaded to https://doi.org/10.5061/dryad.gqnk98sn4.

The following datasets were generated:

| Author(s) | Year | Dataset title | Dataset URL | Database and Identifier |
|---|---|---|---|---|
| Goldman RD | 2021 | Structural heterogeneity of cellular K5/K14 filaments as revealed by cryo-electron microscopy | https://doi.org/10.5061/dryad.gqnk98sn4 | Dryad Digital Repository, 10.5061/dryad.gqnk98sn4 |
| Weber MS, Eibauer M, Medalia O | 2021 | Cryo-tomogram of the K5/K14 keratin network in a keratinocyte ghost cell | https://www.ebi.ac.uk/emdb/entry/EMD-12959 | EMBD, EMD-12959 |

Weber MS, Eibauer M, Medalia O  2021  Cryo-tomogram showing the modulation of the K5/K14 keratin network in a keratinocyte ghost cell  https://www.ebi.ac.uk/emdb/entry/EMD-12958  EMBD, EMD-12958

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
