## [Decision Letter]

**Acceptance summary:**

In this study, Weber et al. study the structure of keratin intermediate filaments (KIFs) in detergent extracted 'ghost' cells using cryo-EM. To limit compositional heterogeneity the authors generate a cell line which contains only K5/K14 filaments. They show the KIFs vary in diameter and protofilament helicity. The authors also use cryo-electron tomography to show that KIFs contain density in their center. In some cases, six protofilaments are clearly visible in the cross sections but there also appears to be heterogeneity in their number. This data is a first in the field of keratin research, and provides a much needed framework to understand the structure, properties, and roles of keratin filaments.

**Decision letter after peer review:**

Thank you for submitting your article "Structural heterogeneity of cellular K5/K14 filaments as revealed by cryo- electron microscopy" for consideration by *eLife*. Your article has been reviewed by 2 peer reviewers, and the evaluation has been overseen by a Reviewing Editor and Anna Akhmanova as the Senior Editor. The following individual involved in review of your submission has agreed to reveal their identity: Helen Foster (Reviewer #2).

The reviewers have discussed their reviews with one another, and the Reviewing Editor has drafted this to help you prepare a revised submission. The reviewers agreed that the work is an important step forward for the keratin field and is well executed. They suggest only textual revisions. Please see their full reviews below.

Essential Revisions (for the authors):

1) The discussion of the variable length repeating patterns is not currently clear. In the results (p. 9 and 12), the authors conclude that the observed variability is due to change in Z-height of filament and associated tilt of the filament along its axis. However, the differences in helical pattern are included as a feature of protofilament heterogeneity in the discussion (line 382). Based on the current data, the differences detected could be due to changes in Z-height or true variation in protofilament arrangement. Ideally this should be clarified through classification of 3D sub volumes of the cryo-ET data but this will be technically challenging. I suggest the authors conclude both options are possible in the results and limit the discussion of protofilament arrangement to differences in straight vs. twisted, which are clear in the data.

2) How rigorous are the biochemical analyses that were conducted to establish that K5 is the only type II keratin expressed in the recombinant mouse keratinocytes? Also, have the authors effectively ruled out that other IFs, e.g., vimentin (nestin?), could be expressed in some of the keratinocytes analyzed?

3) How many keratinocytes have been sampled to constitute the bank of images subjected to processing and analysis? What criteria were considered when opting to include/exclude cells from the analysis?

4) Figure 1 relates that the sampling of filaments for analysis was taken at the periphery of the cell, as expected. This is where keratin filament assembly takes place, certainly in cultured cells, as originally shown by Rudolf Leube in 2004 (MBoC) and confirmed for additional cell types including mouse skin keratinocytes in primary culture (Feng et al., J Cell Biol 2015). Therefore, it may be that the heterogeneity in filament structure reflect, to a degree, an immature or yet to be completed assembly state for the fibers. Another potential determinant of the hypervariable structure of the fibers may stem from the use of detergents to prepare cell ghosts. Yet another contributing factor may be that individual/isolated cells were included in the study, whereas keratinocytes are naturally occurring as multicellular assemblies. Surprisingly, these potential factors are not discussed in the manuscript – at least, not as much as they should.

5) Aside from the role of tilting adeptly described by the authors, could there be a relationship between the size of the repeat distance of the axial helical pattern and filament diameter? Related to this, could the filaments or filament segments showing a longer repeat distance be "stretched", or those with a shorter repeat distance be "compressed"? This is a very interesting aspect of the study.

6) The authors should relate the caveats and/or assumptions associated with the approach they used to "reconstitute" the filaments (lines 227-240). I thought that as actually reconstituted filaments from purified K5 and K14 proteins – including that reference would have been very exciting.

7) The variable sub-architecture of filaments may be a key determinant that underlies the property of self-organization of keratin filaments – the latter has been best characterized, biochemically, in a 2002 MBoC paper by Yamada et al. 2009 J Cell Biol paper by Lee and Coulombe. Upon a modest shift in pH (along physiological lines; see Matsui et al., PNAS 2021), K5-K14 undergo bundling to a remarkable degree, a property the significantly enhances their micromechanical properties (cf. Ma et al., Nat Cel Biol 2001; Yamada et al. MBoC 2002). Please consider discussing.

8) Lines 351-352. Given all the biases that have gone into selection of individual fibers for analysis, I question the rationale for spending so much time discussing persistence length, given that in this regard this current study contributes little to what was already known in that regard.

9) The observation of a "central core", if this is real, is a game changer in the field. Could the core density be "just another protofilament" that happens the be centrally located in cross-section? Does this core show a helical pattern along the main axis of the fiber? Is the core present whether the filament are comprised of protofilaments showing a heliocal pattern, or only in filaments for which the protofilaments are "straight? Finally, could the core be related to the end domains? (unlikely, but…) Here, it is important to recognize that Lars Norlen (J Invest Dermatol 2004) has reported on such a central core in native keratin filaments in cross-section when analyzing terminally differentiated keratinocytes of epidermis using cryoEM analysis of sections prepared from flash-frozen, vitrified skin tissue. His findings should be discussed.

10) What about the unstructured end domains at the N- and C-termini of K5 and K14? Any insight about where they may be located? The head domains, in particular, are as "large" in primary structure as are subdomain 1B and coil 2.

11) Please comment on how this work compares to two earlier reports in the literature. I.e. the cryo-EM of vimentin filaments (Goldie et al. JSB 2007), that shows an octameric, i.e. protofibrillar organization of IF, in agreement with the keratin work published by Ueli Aebi (JCB 1983). In the latter paper, a four-stranded helical arrangement of protofibrils is shown. Also, the assembly process via unit-length filaments gives a symmetry of filaments according to their repeat-length of 43 nm, etc.

*Reviewer #1 (Recommendations for the authors):*

Main issues to consider

How rigorous are the biochemical analyses that were conducted to establish that K5 is the only type II keratin expressed in the recombinant mouse keratinocytes? Also, have the authors effectively ruled out that other IFs, e.g., vimentin (nestin?), could be expressed in some of the keratinocytes analyzed? Given what is at stake here, might it not be desirable to conduct a MS-based survey of all the proteins contained in the cell ghosts subjected to cryoET analyses?

How many keratinocytes have been sampled to constitute the bank of images subjected to processing and analysis? What criteria were considered when opting to include/exclude cells from the analysis?

Figure 1 relates that the sampling of filaments for analysis was taken at the periphery of the cell, as expected. This is where keratin filament assembly takes place, certainly in cultured cells, as originally shown by Rudolf Leube in 2004 (MBoC) and confirmed for additional cell types including mouse skin keratinocytes in primary culture (Feng et al., J Cell Biol 2015). Therefore, it may be that the heterogeneity in filament structure reflect, to a degree, an immature or yet to be completed assembly state for the fibers. Another potential determinant of the hypervariable structure of the fibers may stem from the use of detergents to prepare cell ghosts. Yet another contributing factor may be that individual/isolated cells were included in the study, whereas keratinocytes are naturally occurring as multicellular assemblies. Surprisingly, these potential factors are not discussed in the manuscript – at least, not as much as they should.

Aside from the role of tilting adeptly described by the authors, could there be a relationship between the size of the repeat distance of the axial helical pattern and filament diameter? Related to this, could the filaments or filament segments showing a longer repeat distance be "stretched", or those with a shorter repeat distance be "compressed"? This is a very interesting aspect of the study.

The authors should relate the caveats and/or assumptions associated with the approach they used to "reconstitute" the filaments (lines 227-240). I thought that has actually reconstituted filaments from purified K5 and K14 proteins – including that reference would have been very exciting.

The variable sub-architecture of filaments may be a key determinant that underlies the property of self-organization of keratin filaments – the latter has been best characterized, biochemically, in a 2002 MBoC paper by Yamada et al., 2009 J Cell Biol paper by Lee and Coulombe. Upon a modest shift in pH (along physiological lines; see Matsui et al., PNAS 2021), K5-K14 undergo bundling to a remarkable degree, a property the significantly enhances their micromechanical properties (cf. Ma et al., Nat Cel Biol 2001; Yamada et al., MBoC 2002).

Lines 351-352. Given all the biases that have gone into selection of individual fibers for analysis, I question the rationale for spending so much time discussing persistence length, given that in this regard this current study contributes little to what was already known in that regard.

The observation of a "central core", if this is real, is a game changer in the field. Could the core density be "just another protofilament" that happens the be centrally located in cross-section? Does this core show a helical pattern along the main axis of the fiber? Is the core present whether the filament are comprised of protofilaments showing a heliocal pattern, or only in filaments for which the protofilaments are "straight? Finally, could the core be related to the end domains? (unlikely, but…) Here, it is important to recognize that Lars Norlen (J Invest Dermatol 2004) has reported on such a central core in native keratin filaments in cross-section when analyzing terminally differentiated keratinocytes of epidermis using cryoEM analysis of sections prepared from flash-frozen, vitrified skin tissue. His findings should be discussed.

What about the unstructured end domains at the N- and C-termini of K5 and K14? Any insight about where they may be located? The head domains, in particular, are as "large" in primary structure as are subdomain 1B and coil 2.

*Reviewer #2 (Recommendations for the authors):*

This paper clearly shows that cellular keratin intermediate filaments are highly heterogenous. I only have one main point for the authors to clarify.

Main point:

– The discussion of the variable length repeating patterns is not currently clear. In the results (p. 9 and 12), the authors conclude that the observed variability is due to change in Z-height of filament and associated tilt of the filament along its axis. However, the differences in helical pattern are included as a feature of protofilament heterogeneity in the discussion (line 382). Based on the current data, the differences detected could be due to changes in Z-height or true variation in protofilament arrangement. Ideally this should be clarified through classification of 3D sub volumes of the cryo-ET data but this will be technically challenging. I suggest the authors conclude both options are possible in the results and limit the discussion of protofilament arrangement to differences in straight vs. twisted, which are clear in the data.

---

## [Author Response]

Essential Revisions (for the authors):1) The discussion of the variable length repeating patterns is not currently clear. In the results (p. 9 and 12), the authors conclude that the observed variability is due to change in Z-height of filament and associated tilt of the filament along its axis. However, the differences in helical pattern are included as a feature of protofilament heterogeneity in the discussion (line 382). Based on the current data, the differences detected could be due to changes in Z-height or true variation in protofilament arrangement. Ideally this should be clarified through classification of 3D sub volumes of the cryo-ET data but this will be technically challenging. I suggest the authors conclude both options are possible in the results and limit the discussion of protofilament arrangement to differences in straight vs. twisted, which are clear in the data.

We thank the Reviewer for this comment. Indeed, the variations detected in the repeating pattern could be attributed both to filament tilting or/and to variations in protofilament arrangement. In the revised manuscript, we emphasize this point in the results and Discussion sections, where we focus on the obvious heterogeneity of keratin filaments, i.e., variability in diameter, alternating helical and straight filament patterns.

2) How rigorous are the biochemical analyses that were conducted to establish that K5 is the only type II keratin expressed in the recombinant mouse keratinocytes? Also, have the authors effectively ruled out that other IFs, e.g., vimentin (nestin?), could be expressed in some of the keratinocytes analyzed?

The cell line we used for the Krt6a and Krt6b knockout is a well-characterized and authenticated cell line, generated in the Magin lab, termed KtyI KO K14 cell line. The KtyI KO keratinocytes were extracted from knockout mice lacking the entire type I keratin cluster. After lentiviral transduction of the Krt14 gene, these KtyI KO K14 keratinocytes were able to re-form a keratin cytoskeleton meshwork, consisting of K5/K14 filaments and K6/K14 filaments. Expression of non-keratin IFs, e.g. vimentin, was excluded by analyses performed in the Magin lab. In the KtyI deficient embryos, vimentin expression was excluded by RT-PCR, in keratinocytes, illumina arrays were performed that excluded the expression of non-keratin IFs (References: Kumar et al., 2016, J Allergy Clin Immunol; Kumar et al., 2015, J Cell Biol). These references are included and we have now clarified this point in the revised manuscript.

These analyses showed that K5, K6a, K6b and K14 are the only intermediate filaments expressed in KtyI KO K14 cells. After the CRISPR/Cas9 knockout of the Krt6a and Krt6b gene, the knockout was verified by genomic DNA analysis of the Krt6a and Krt6b genes, which revealed frame-shift mutations (Figure 1—figure supplement 1A, B). This produces a homogenous K6a and a heterogenous K6b knockout. However, immunostaining revealed that no K6b/K14 filaments are assembled (Figure 1—figure supplement 1C). We could not rule out that non-filamentous K6b protein is present in the K5/K14_1 cells, but it does not assemble into filaments and therefore does not interfere with our analysis. Furthermore, non-filamentous soluble keratin proteins would be removed during the cell permeabilization step and are therefore not present in the ghost cells. It can be concluded that K5/K14 filaments are the only IFs present in the sample analyzed for this study. We highlighted this in the Results section.

3) How many keratinocytes have been sampled to constitute the bank of images subjected to processing and analysis? What criteria were considered when opting to include/exclude cells from the analysis?

For the different datasets, micrographs were collected on a total of ~ 60 ghost cells. The dataset at 22,655x, which was used for most experiments including 2D classifications and computational reconstitution of long keratin filaments was acquired on ~ 20 ghost cells.

No cell that was used for data collection was excluded at any later stage of analysis. Data were recorded at positions where the thickness of the ghost cells allowed cryo-EM analysis. Overview images of entire ghost cells were acquired prior to data acquisition, in which the filaments were identified. Individual micrographs were excluded if they contained no keratin filaments or if the cellular environment was so densely packed that no individual keratin filaments could be identified. Additionally, only high-quality images were used, assessed by CTF estimation (CTF information exceeds 7.5 Å).

We included the information above in the Methods section.

4) Figure 1 relates that the sampling of filaments for analysis was taken at the periphery of the cell, as expected. This is where keratin filament assembly takes place, certainly in cultured cells, as originally shown by Rudolf Leube in 2004 (MBoC) and confirmed for additional cell types including mouse skin keratinocytes in primary culture (Feng et al., J Cell Biol 2015). Therefore, it may be that the heterogeneity in filament structure reflect, to a degree, an immature or yet to be completed assembly state for the fibers. Another potential determinant of the hypervariable structure of the fibers may stem from the use of detergents to prepare cell ghosts. Yet another contributing factor may be that individual/isolated cells were included in the study, whereas keratinocytes are naturally occurring as multicellular assemblies. Surprisingly, these potential factors are not discussed in the manuscript – at least, not as much as they should.

We thank the Reviewer for this comment.

Although much of the data was acquired close to the cell periphery, most images were acquired several microns into the cells, while other micrographs were collected deeper within the cell; sometimes in close proximity to the nucleus. Thus, our data was not only acquired close to the plasma membrane, but reflects keratin filaments from most parts of the cell. Further, filaments with obvious ends were excluded from our analysis. This is now clarified in the Discussion section.

We showed that the structure of long, single filaments is modulated along their long axis, and the observed heterogeneity occurred not only between, but especially within individual filaments. It is unlikely that all of the 4,460 filaments that we reconstituted in silico (Figure 3 and Figure 3 supplement 1) are not fully assembled, especially since some were found close to the cell nucleus.

Therefore, we believe that the changes in structure reflect an intrinsic property of keratin IFs throughout the cell. However, it may be that our structural analysis includes a subset of filaments in different states of assembly, but we did not detect a unique structural motif that stood out in any subcellular location. However, in light of this comment, we discuss this possibility in the Discussion section.

Traditionally, structural biology relies on successful purification which involves multiple steps, some of which may include detergents or other chemicals. Here we have minimized this procedure down to a matter of minutes in order to reduce the risk of structural changes. The diluted detergent (0.1 % Triton X-100) was used for 15-20 seconds, to prepare ghost cells which contained typical keratin filament networks. To assess the effect of the detergent treatment on protein structures we analyzed actin filaments from the same dataset and could show that the actin structure is unaltered and can be resolved to 6.1 Å (Figure 1—figure supplement 2). Moreover, previous studies on lamins, a type V intermediate filament, showed, that the structure of lamin filaments is not altered after a similar detergent treatment when compared to cryo-FIB milling and tomography of unextracted cells (Kronenberg-Tenga et al., 2021). Therefore, we have concluded that it is unlikely that the detergent treatment alters the keratin IF structure significantly. In line with this reviewer’s comment, we elaborate on this in the Discussion section.

In our experiments, cells were not studied in isolation. Keratinocytes require close associations with neighboring cells, so they were always cultured at high densities (minimum of 50%) on EM grids. Typically, in our preparations cells are directly attached to other cells (for example, a neighboring cell can be seen in Figure 1B). Nevertheless, it may be possible that the KIF network organization in the cultured keratinocytes is different than in keratinocytes which are grown in denser monolayers or in tissues. Therefore we do not draw conclusions about the keratin IF network organization. With this approach it is unlikely that the structure of individual keratin IF themselves is altered.

5) Aside from the role of tilting adeptly described by the authors, could there be a relationship between the size of the repeat distance of the axial helical pattern and filament diameter? Related to this, could the filaments or filament segments showing a longer repeat distance be "stretched", or those with a shorter repeat distance be "compressed"? This is a very interesting aspect of the study.

This is a very interesting point. We re-analyzed our data and compared the diameters of 2D classes with differences in the repeating pattern, but we could not find a correlation between the diameter and the length of the repeating pattern. Actually, the 2D classes which show a helical repeating pattern are all ~ 10 nm in diameter. The very wide (~13 nm) or very thin (~9.2 nm) classes do not show a clear repeating pattern, so the comparison is not straightforward.

It is still possible that the filament segments with longer repeat distances are “stretched” and those with shorter repeat distances are “compressed”. However, we believe that this is unlikely, since we studied the keratin IFs after releasing the cellular tension that was applied to the filaments (cells were lysed). Thus, it seems likely that only irreversible stretching should be retained in our analysis. Irreversible stretching could be induced by extensive stretching of filaments above 200%, which would result in α-helix to β-sheet transitions of the coiled-coil domains (Kreplak et al., 2005). The 2D classes with very thin diameters and a straight appearance might reflect such irreversible stretched filaments. Since these classes are frequently found in the cellular assembled keratins, this may be physiologically relevant.

Tilting seems to be the most likely explanation for the differences in the repeating distance, because we already see in the tomograms that tilted filaments are present in the sample.

We added the above to the Discussion section.

6) The authors should relate the caveats and/or assumptions associated with the approach they used to "reconstitute" the filaments (lines 227-240). I thought that as actually reconstituted filaments from purified K5 and K14 proteins – including that reference would have been very exciting.

We apologize for the misunderstanding. In this study, we only used a computational in silico reconstitution. Namely, we used the 2D class averaging and mapped the classes back to the original position from which the segments originated along the filaments. This approach was developed in Martins et al., 2021. Throughout the entire study we used only cellular assembled K5/K14 filaments and studied them only within ghost cells. We revised the explanation and changed the term into ‘computational reconstitution’.

7) The variable sub-architecture of filaments may be a key determinant that underlies the property of self-organization of keratin filaments – the latter has been best characterized, biochemically, in a 2002 MBoC paper by Yamada et al. 2009 J Cell Biol paper by Lee and Coulombe. Upon a modest shift in pH (along physiological lines; see Matsui et al., PNAS 2021), K5-K14 undergo bundling to a remarkable degree, a property the significantly enhances their micromechanical properties (cf. Ma et al., Nat Cel Biol 2001; Yamada et al. MBoC 2002). Please consider discussing.

We thank the reviewers for this point. To our knowledge, Lee and Coulombe, 2009 and Yamada et al., 2002, focused on the bundling properties of keratin filaments and the regions involved in inter-filament interactions. This is very interesting and their studies yielded fascinating insights into the self-organizing properties of K5/K14 filaments. In our analysis, we focused on single individual filaments, non-bundled filaments, and applied single particle image processing procedures. Furthermore, the sample preparation procedure involved the removal of soluble parts of the cytoplasm and therefore some changes in the keratin IF network may occur. Since we cannot guarantee that the network organization we observe in our ghost cells is the same as it was in intact cells, we only focused on individual keratin IFs. We have now referred to these articles in the context of keratin meshworks.

8) Lines 351-352. Given all the biases that have gone into selection of individual fibers for analysis, I question the rationale for spending so much time discussing persistence length, given that in this regard this current study contributes little to what was already known in that regard.

We thank the Reviewer for this comment and agree that the discussion about the persistence length is very long. We have shortened this section.

9) The observation of a "central core", if this is real, is a game changer in the field. Could the core density be "just another protofilament" that happens the be centrally located in cross-section? Does this core show a helical pattern along the main axis of the fiber? Is the core present whether the filament are comprised of protofilaments showing a heliocal pattern, or only in filaments for which the protofilaments are "straight? Finally, could the core be related to the end domains? (unlikely, but…) Here, it is important to recognize that Lars Norlen (J Invest Dermatol 2004) has reported on such a central core in native keratin filaments in cross-section when analyzing terminally differentiated keratinocytes of epidermis using cryoEM analysis of sections prepared from flash-frozen, vitrified skin tissue. His findings should be discussed.

The central core density may represent an additional protofilament that is located in the center of the filament. It was found in all the keratin IFs analyzed in this study, regardless of whether they show a straight or a helical pattern. As a control, we calculated a rotationally averaged structure as seen in Figure 4G separately from only helical and only straight 2D classes and both showed a central density. Furthermore, no cross-sectional view could be detected where the filament central core was missing. In the revised manuscript we further analyzed filaments seen in cryo-tomograms (Figure 4—figure supplement 1E), it showed a continuous density within the luminal space. It further supports the idea that the density is of structured domains, rather than the unstructured head or tail domains.

Moreover, we recently identified a central luminal density in vimentin filaments, and verified that it cannot be the head or the tail of vimentin (Eibauer et al., 2021 BioRxiv). Although major structural differences between keratin and vimentin IFs are apparent, they share some common structural properties.

Whether the central density follows a helical symmetry remains unknown. However, due to steric restriction, it is unlikely that it would exhibit the same helical parameters as the tubular keratin scaffold. In the additional analysis that was added to the revised manuscript, we show that the central density is not always located in the exact center of the filament, but can move within the filament tube (Figure 4—figure supplement 1E). These results are presented and discussed in the revised manuscript.

We thank the Reviewers for pointing out the studies by Norlen et al. Their findings agree very well with our findings and are discussed in the revised manuscript.

10) What about the unstructured end domains at the N- and C-termini of K5 and K14? Any insight about where they may be located? The head domains, in particular, are as "large" in primary structure as are subdomain 1B and coil 2.

The flexible head and tail domains were predicted to be intrinsically unstructured. Therefore, it is likely that they would not have a substantial contribution to the densities of the averaged KIF structure. A high-resolution 3D structural analysis would be necessary to answer these questions. Unfortunately, we are not sure we can deduce that information from our vimentin IF structural analysis since they clearly assemble differently and a minor rotation of keratin protofilaments may result in different directions of the N- and C- termini.

11) Please comment on how this work compares to two earlier reports in the literature. I.e. the cryo-EM of vimentin filaments (Goldie et al., JSB 2007), that shows an octameric, i.e. protofibrillar organization of IF, in agreement with the keratin work published by Ueli Aebi (JCB 1983). In the latter paper, a four-stranded helical arrangement of protofibrils is shown. Also, the assembly process via unit-length filaments gives a symmetry of filaments according to their repeat-length of 43 nm, etc.

We thank the Reviewer for this comment and have now added a paragraph in the discussion regarding these references.

As mentioned above, we see major structural differences between vimentin and keratin IFs. Although some literature proposed the idea of universal 32-mer IFs, which would be the basic building block of all cytoplasmic intermediate filaments, this model never agreed with the experimental evidence. Keratin IFs contain much less mass per length (Engel et al., 1985) than vimentin (Herrmann et al., 1996) and therefore fewer monomers in cross-section. Therefore, at this stage we are not confident that vimentin and keratin filaments are similar.

Our study supports the notion that IFs do not adhere to a common building plan, but show substantial diversity among the different types (Goldie et al., 2007).

We thank the reviewers for pointing out that the ‘repeat distance’ and of a helix should be clarified, as it might cause confusion with the ‘repeating length’ of a helix, which is not the parameter we measured here. In helical parameter terms, our ‘repeat distance’ refers to the ‘pitch’ of a helix, describing the distance required to complete a 360° helix turn. To clarify the helical parameter we determined with the autocorrelation analysis, we renamed the repeat distance to ‘pitch’, as this represents the correct nomenclature. Our measurements therefore describe different structural features than the 43 nm repeat length of an ULF. It agrees with the helical pitch shown in Aebi et al., 1983, measured by Power spectrum analysis of negatively stained keratin filaments.